# Breakdown of semiclassical description of thermoelectricity in near-magic angle twisted bilayer graphene

Bhaskar Ghawri [1✉], Phanibhusan S. Mahapatra[1✉], Manjari Garg [2✉], Shinjan Mandal [1], Saisab Bhowmik[2], Aditya Jayaraman [1], Radhika Soni[2], Kenji Watanabe [3], Takashi Taniguchi [4], H. R. Krishnamurthy[1], Manish Jain [1], Sumilan Banerjee [1], U. Chandni [2] & Arindam Ghosh[1,5✉]

The planar assembly of twisted bilayer graphene (tBLG) hosts multitude of interaction-driven phases when the relative rotation is close to the magic angle ($\theta_m = 1.1°$). This includes correlation-induced ground states that reveal spontaneous symmetry breaking at low temperature, as well as possibility of non-Fermi liquid (NFL) excitations. However, experimentally, manifestation of NFL effects in transport properties of twisted bilayer graphene remains ambiguous. Here we report simultaneous measurements of electrical resistivity ($\rho$) and thermoelectric power ($S$) in tBLG for several twist angles between $\theta \sim 1.0 - 1.7°$. We observe an emergent violation of the semiclassical Mott relation in the form of excess $S$ close to half-filling for $\theta \sim 1.6°$ that vanishes for $\theta \gtrsim 2°$. The excess $S$ ($\approx 2\,\mu$V/K at low temperatures $T \sim 10$ K at $\theta \approx 1.6°$) persists upto $\approx 40$ K, and is accompanied by metallic $T$-linear $\rho$ with transport scattering rate ($\tau^{-1}$) of near-Planckian magnitude $\tau^{-1} \sim k_B T/\hbar$. Closer to $\theta_m$, the excess $S$ was also observed for fractional band filling ($\nu \approx 0.5$). The combination of non-trivial electrical transport and violation of Mott relation provides compelling evidence of NFL physics intrinsic to tBLG.

[1] Department of Physics, Indian Institute of Science, Bangalore 560012, India. [2] Department of Instrumentation and Applied Physics, Indian Institute of Science, Bangalore 560012, India. [3] Research Center for Functional Materials, National Institute for Materials Science, Namiki 1-1, Tsukuba, Ibaraki 305-0044, Japan. [4] International Center for Materials Nanoarchitectonics, National Institute for Materials Science, Namiki 1-1, Tsukuba, Ibaraki 305-0044, Japan. [5] Centre for Nano Science and Engineering, Indian Institute of Science, Bangalore 560 012, India. ✉email: gbhaskar@iisc.ac.in; phanis@iisc.ac.in; manjarigarg@iisc.ac.in; arindam@iisc.ac.in

In moiré systems with twisted bilayer graphene (tBLG), the amplification of Coulomb correlation effects at low twist angles ($\theta$) is a result of nearly flat low-energy electronic bands[1,2] and divergent density of states (DOS) at van Hove singularities (vHS)[3]. In addition to superconductivity[4], ferromagnetism[5], the strong correlation effects in tBLG manifest in a cascade of broken symmetry phases at integer band filling factor ($\nu$) close to $\theta = \theta_{\mathrm{m}}$[6,7]. Near half-filling ($\nu = \pm 2$) of the four-fold spin-valley degenerate conduction and valence bands, a linear $T$-dependence of the resistivity ($\rho$) seems to indicate an interaction-related absence of a well-defined quasiparticle spectrum, which is concomitant with non-Fermi liquid (NFL) excitations[8,9]. The persistence of the linearity in $\rho$ for $\theta$ well away from $\theta_{\mathrm{m}}$, e.g. for $\theta \sim 1.5 - 2°$, however, has been interpreted in terms of a contrary scenario that is addressable within the non-interacting framework[10]. The uncertainty persists even in scanning tunneling microscopy experiments[11–13], where the possibility of an interaction-driven magnetic order has been claimed close to the vHS for $\theta$ as high as 1.6°, although the spontaneous breaking of $C_6$ lattice symmetry to nematic orbital order has not been observed for $\theta > \theta_{\mathrm{m}}$. Thus a comprehensive understanding of the impact of correlation in tBLG requires a complementary experimental probe that is capable of identifying the departure from non interacting physics in an unambiguous manner.

Here we have carried out simultaneous electrical and thermoelectric measurements in tBLG for twist angles varying from $\theta \sim 1.0 - 1.7°$. The dependence on $T$ and on the carrier density ($n$) of the thermoelectric power ($S$), or the Seebeck coefficient, is used as an independent and sensitive probe of the correlation effects. Thermoelectric power is often interpreted as a thermodynamic entity that represents the entropy carried by each charge carrier. Within the degenerate quasiparticle description in the Boltzmann transport regime ($T \ll T_{\mathrm{F}}$, where $T_{\mathrm{F}}$ is the Fermi

temperature), $S$ is related to the resistance ($R$) through the semiclassical Mott relation (SMR),

$$S_{\mathrm{Mott}} = \frac{\pi^2 k_{\mathrm{B}}^2 T}{3|e|} \frac{\mathrm{d}\ln R(E)}{\mathrm{d}E}\Bigg|_{E_{\mathrm{F}}}, \qquad (1)$$

where $R(E)$, $e$ and $E_{\mathrm{F}}$ are the energy-dependent resistance, electronic charge and Fermi energy, respectively. Eq. (1) is valid for a quasi-particle description of transport using semiclassical Boltzmann equation under the assumption that scattering is elastic close to Fermi surface. Remarkably, this simple assumption of elastic quasiparticle scattering remain valid in a wide variety of systems, such as disordered metals/semiconductors[14,15], organic materials[16], monolayer graphene[17] and topological insulators[18]. The SMR effectively arises from the quasiparticles carrying heat and charge under identical constraints, imposed by the momentum conservation. Thus, the validity of SMR in Eq. (1) provides a definitive probe into the nature of the scattering mechanisms and energy distribution of the charge carriers near the Fermi surface, and it breaks down when strong correlation effects become important[15,19].

## Results

The tBLG devices we study were created using standard van der Waals stacking[20], which consists of two graphene layers aligned at either $60° + \theta$ or at $\theta$, where $\theta$ is the effective twist angle, and encapsulated within two sheets of hexagonal boron nitride (hBN) (See Supplementary Note 1). A local top-gate tunes $n$ in the overlap region where the moiré super-lattice is formed. Figure 1a shows the four terminal resistance $R$ measured across the tBLG devices as a function of band filling factor $\nu$ and $T$ for four different $\theta$. The recurring features in $R$ across the tBLG devices can be identified as the maxima in $R$ at the charge neutrality point (CNP) and at the full-filling of the moiré band ($\nu = \pm 4$).

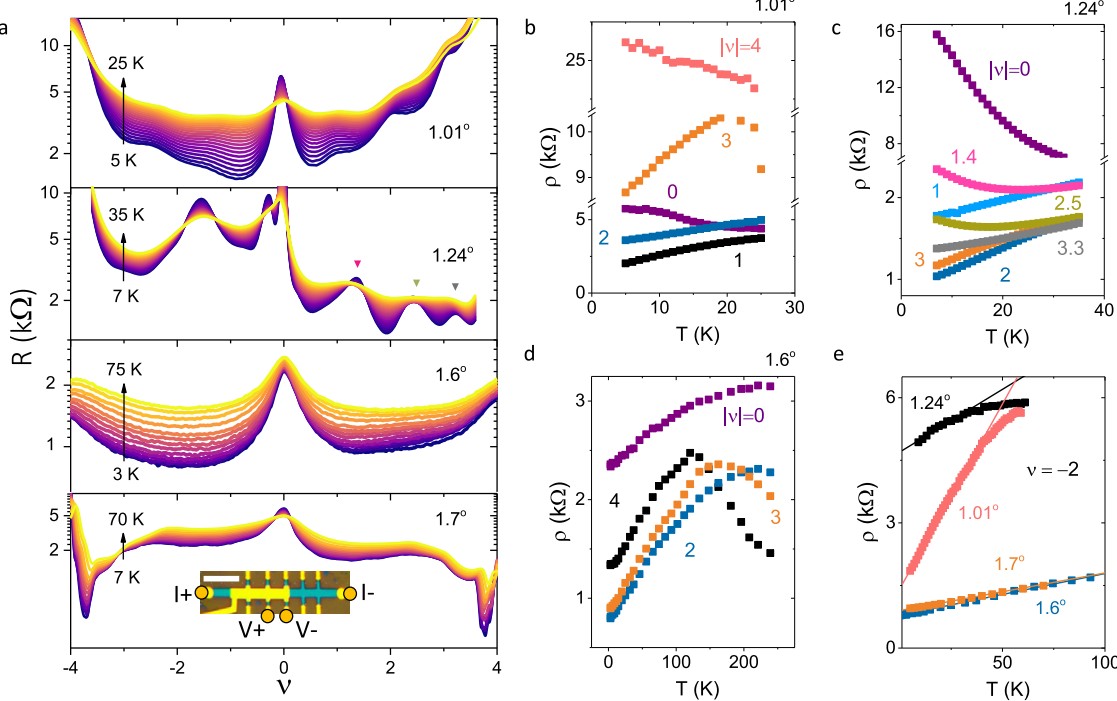

**Fig. 1 Electrical transport in twisted bilayer graphene. a** Temperature-dependent resistance $R$ as a function of band filling $\nu$ for four different devices with twist angle $\theta \approx 1.01°$, 1.24°, 1.6° and 1.7°. The inset in the bottom panel shows the optical image of a typical device with current and voltage leads marked for resistivity measurements. The scale bar represents a length of 5 μm. $\rho$ as a function of $T$ for a few representative values of $\nu$ for (**b**) $\theta \approx 1.01°$, (**c**) 1.24° and (**d**) 1.6°. The various curves for fractional $\nu$ in (**c**) represent the $T$-dependence of the CI/CS states as marked in the second panel of (**a**). **e** Comparison of $T$-dependence of $\rho$ at $\nu = -2$ for different twist angles. The solid lines represent $T$-linearity.

In addition, near $\theta_m$, the device with $\theta \approx 1.01°$ exhibit additional maxima in $R$ at integer values of $\nu$, whereas for $\theta \approx 1.24°$ resistance peaks are shifted slightly away from integer fillings. For $\theta \approx 1.24°$, we observe a substantial shift $|\Delta\nu| \sim 0.25$ of resistance peaks near $\nu = +1$ and $+3$ from 7 K to 35 K, suggesting the possibility of isospin-polarization in the system[21,22] (See Supplementary Note 4). We speculate that the noticeable asymmetry in the doping dependence of $R$ on the electron and hole sides is most likely related to the particle-hole asymmetry of the band structure since in both tBLG devices near $\theta_m$, the correlated states are more pronounced at electron doping.

For $0 < |\nu| < 4$, the $T$-dependence for all devices was found to be generally metallic at low temperatures $\lesssim 40$ K (Fig. 1b–d). However, the resistance peaks near integer $\nu$ exhibit either weak insulating $T$-dependence, i.e, a correlated insulating phase (CI), or $T$-linear resistivity i.e, a correlated semimetallic (CS) phase. In the metallic regime, $\rho$ can expressed as $\rho = \rho_0 + AT$, where $\rho_0$ is the residual resistivity. The values of $A$ ($\sim 10$–$100$ $\Omega$/K) are at least two orders of magnitude greater compared to that of the monolayer graphene, and are consistent with the earlier transport measurements in tBLG devices[10]. From the comparison of $\rho(T)$ at half-filling ($\nu = -2$) in Fig. 1e, we find that the resistivity is linear in $T$ for all four low-angle devices. However, at $\theta \approx 1.24°$ and $\theta \approx 1.01°$, the linearity persists only upto $\sim 40$ K, which could be due to the smaller bandwidth and smaller bandgap[10]. We also note that the value of $A$ ($\sim 40$–$100$ $\Omega$/K) at $\nu = -2$ is much larger near $\theta \sim \theta_m$ than that of the devices away from $\theta_m$ (See Supplementary Note 4). The ubiquitous $T$-linearity across all tBLG devices at low temperature is a clear departure from $\rho \sim T^2$ dependence associated with electron-electron scattering, or the $\rho \propto T^4$ behavior, expected due to electron-acoustic phonon scattering below the Bloch-Grüneisen temperature ($T_{BG}$)[23]. While this suggests the continuum of correlation-driven metallic states across the tBLG devices, an alterate scenario has been proposed[10,24] to view the tBLG in this regime as a two dimensional, weakly (or non-) interacting metal with largely reduced $T_{BG}$.

To complement the electrical transport, we have performed thermoelectric measurements on the same devices. Briefly, a sinusoidal current ($I_\omega$) is allowed to flow between two contacts of the monolayer branch outside the top gated region, setting up a temperature gradient ($\Delta T$) across the tBLG region (Fig. 2a, b). The resulting second-harmonic thermovoltage ($V_{2\omega}$) is recorded on the tBLG region as a function of doping and heating current[17,20]. The linear response was ensured from $V_{2\omega} \propto I_\omega^2$ for the range of heating current used (Fig. 2c–e). We begin with the results in tBLG devices closer to $\theta_m$. Figure 2c exhibits the $\nu$-dependence of normalized $V_{2\omega}$ for tBLG device $\theta \approx 1.01°$ at low temperature (5 K), which exhibits multiple sign-reversals when $E_F$ is varied across the lowest energy band. While the sign reversals near the CNP and the super-lattice gaps at $\nu = \pm 4$ are due to changes in the quasiparticle excitations, those near integer values of $0 < \nu < 4$ can be attributed to the correlated states. The sign-reversal of $V_{2\omega}$ near each correlated states is fascinating since it indicates a change in the topology of the Fermi surface, which is naturally associated with the Lifshitz transition[25,26]. Although the correlated states are metallic in nature (Fig. 1a), the concomitant Lifshitz transitions depicts the interaction-driven occurrence of diverging DOS at each integer values of $\nu$. This is in stark contrast to the charge-inversions at $\nu = 0, \pm 4$, and hints at the topological facet of the lowest energy band when filled with integer number of charge carriers[7,27].

To establish the connection between the two different types of transports, we rewrite Eq. (1) as,

$$S_{Mott} = \frac{\pi^2 k_B^2 T}{3|e|} \frac{1}{R}\frac{dR}{dV_{tg}}\frac{dV_{tg}}{dn}\frac{dn}{dE}\bigg|_{E_F}, \qquad (2)$$

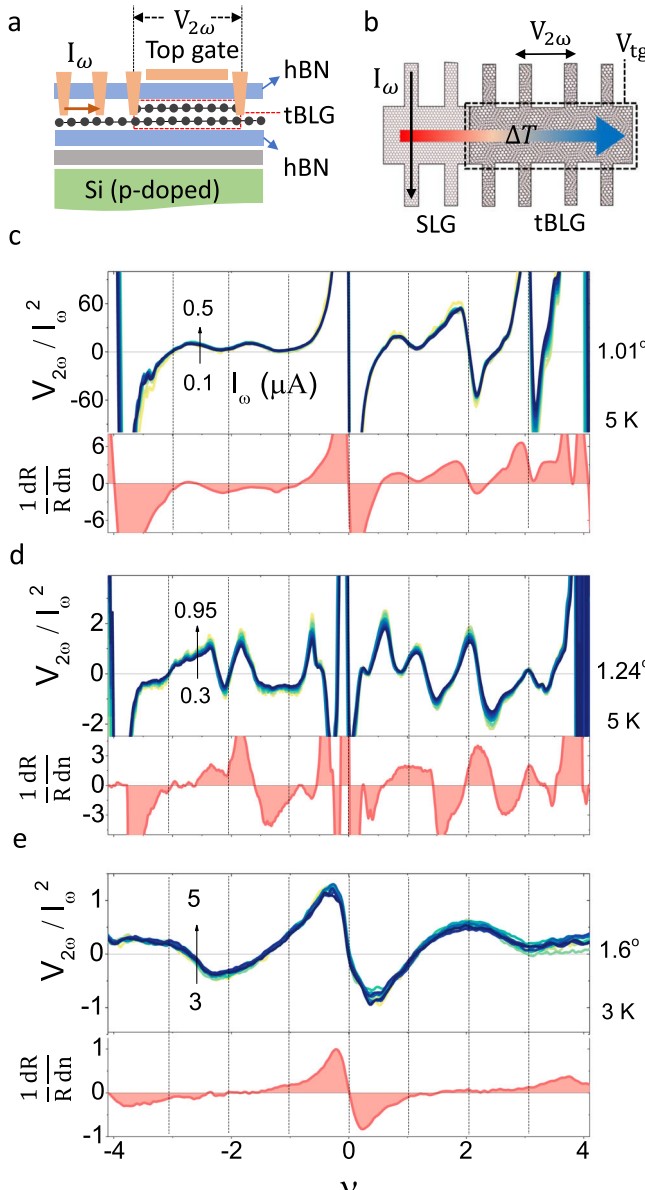

**Fig. 2 Thermoelectric transport in twisted bilayer graphene. a** The cross-sectional view of the device, showing the constituent layers, electrical contacts, and the gate assembly. **b** In-plane heating and measurement schematic for thermovoltage $V_{2\omega}$ in the tBLG region. Density dependence of $V_{2\omega}/I_\omega^2$ (in the units of $VA^{-2} \times 10^6$) measured at 5 K for (**c**) $\theta \approx 1.01°$, **d** 1.24°, and at 3 K for (**e**) 1.6° device. The different curves in each panel represent the $V_{2\omega}/I_\omega^2$ measured at different $I_\omega$. The bottom graphs in each panel show the numerically calculated $\alpha = (1/R)dR/dn$ (in the units of $m^2$) for comparison.

where $(1/R)dR/dV_{tg}$ is measured experimentally, and $dn/dE$ is the DOS ($dV_{tg}/dn = e/C_{hBN}$, where $C_{hBN}$ is the known topgate capacitance per unit area). The difficulty in accurate estimation of DOS in the presence of strong correlation effects near $\theta_m$ prohibits us from accurately estimating $S_{Mott}$, in particular close to the integer fillings for $\theta \approx 1.01°$ and $\approx 1.24°$. Although a qualitative correspondence in the oscillations and sign-reversals of the measured $V_{2\omega}$ and $\alpha = (1/R)dR/dn$ can be seen in these devices including at the CNP and the superlattice gap, absence of accurate knowledge of the DOS prohibits a quantitative estimation of the deviation of the measured thermopower from that expected from the semiclassical model. However, for the device with $\theta \approx 1.24°$

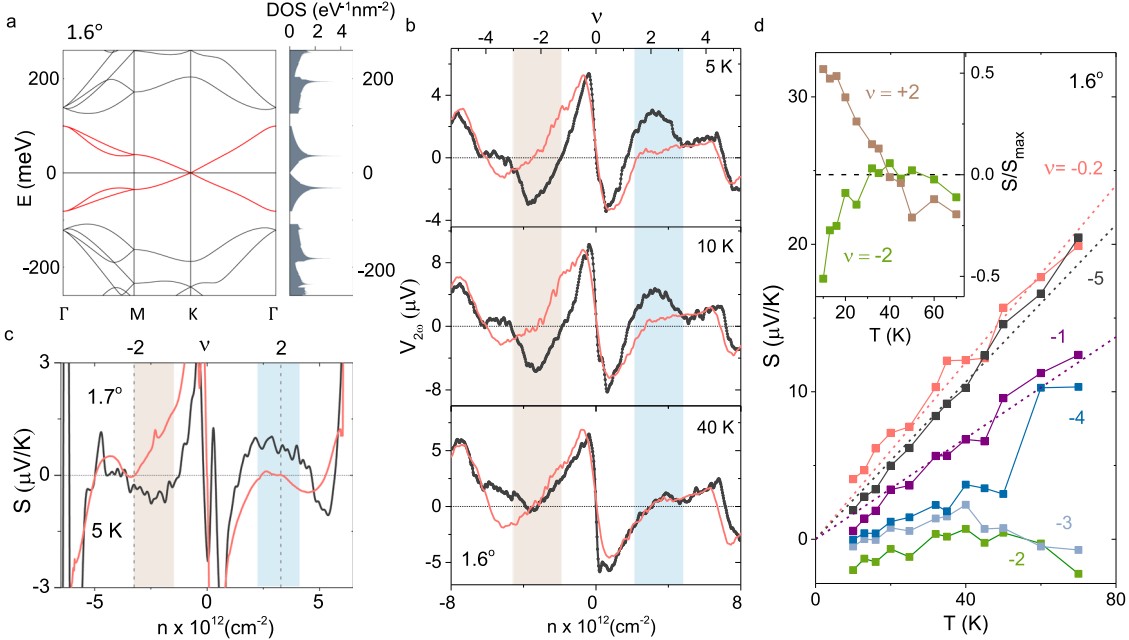

**Fig. 3 Comparison with semiclassical Mott relation at ~1.6°. a** Electronic band structure and density of states (DOS) of tBLG ($\theta = 1.6°$) calculated using tight binding model. The bands shown in red are the low energy active bands. **b** Comparison between the measured $V_{2\omega}$ (black lines) and that calculated (orange line) from the semiclassical Mott relation (Eq. (2)) for $\theta \approx 1.6°$ at three representative temperatures. $\Delta T$ is obtained as a fitting parameter to match SMR with the experimental $V_{2\omega}$ at the CNP. **c** Doping dependence of $S$ for $\theta \approx 1.7°$ compared to that of the SMR at 5 K. **d** Temperature dependence of $S$ at various band filling factors. The dashed lines show the $S \propto T$ dependence. The inset shows the $T$ dependence of $S/S_{max}$ at $\nu = \pm2$.

(Fig. 2d), we detect an excess $V_{2\omega}$ near $\nu \sim 0.5$ which has no analogue in $\alpha$. While this indicates a clear violation of the Mott relation and highlights the possible manifestation of electron-correlation effects at fractional band filling[27], the exact origin of the excess $V_{2\omega}$ at $\nu \approx 0.5$ is not clear at present.

Although the interaction-effects are expected to be weaker when $\theta$ is away from $\theta_m$, the devices $\theta \approx 1.6°$ and $1.7°$ provide a better quantitative comparison with SMR as the non-interacting DOS can be calculated with greater accuracy. The qualitative comparison of $V_{2\omega}$ with $\alpha$ at $\theta \approx 1.6°$ exhibits a discrepancy at low temperature (3 K), where two additional extrema, consisting of a maximum at $\nu = +2$ and minimum at $\nu = -2$, are distinctly absent in $\alpha$ (Fig. 2e). Figure 3a shows the tight binding calculation for the electronic band structure and the corresponding DOS for $\theta \approx 1.6°$ (See Methods and Supplementary Note 8 for more details on the band structure calculations). Using $\Delta T$ as the single fitting parameter, we obtain excellent agreement between the measured $V_{2\omega}$ and Eq. (2) at the CNP ($\nu \sim 0$), $\nu \approx \pm4$, and also in the higher energy dispersive band ($\nu > \pm4$) simultaneously (See Supplementary Fig. 15). For fitting Eq. (2), we also note that $T \ll T_F$ is maintained throughout almost the entire temperature and gate voltage range shown in Fig. 3b, except very close to the CNP ($\nu = 0$) and $\nu = \pm4$ (See Supplementary Note 6). While the SMR explains the observed $V_{2\omega}$ over almost the entire doping regime ($-4 \lesssim \nu \lesssim +4$) at high temperatures ($\gtrsim 40$ K) (bottom panel of Fig. 3b), the excess thermopower centered around $\nu = \pm2$, becomes evident at lower $T$. We also find that the excess thermovoltage is intrinsically particle-hole asymmetric, however, on the electron doped side, the excess thermopower is closer to the commensurate filling ($\nu = +2$) as seen for two devices (Figs. 3b and c). We also detect evidence of small excess $V_{2\omega}$ between $\nu = -3$ and $-4$. This could also be due to electron-correlation effects, however, the exact origin is not clear as we do not observe any evidence of such anomalous thermopower near same filling factor in the other devices (see e.g. Fig. 3c for the device with $\theta \approx 1.7°$). Using the $\Delta T$ extracted from the fitting of $V_{2\omega}$, we show the $T$-dependence of $S = V_{2\omega}/\Delta T$ in Fig. 3d for different $\nu$ (See Supplementary Note 6).

Evidently, $S$ exhibits a linear dependence on $T$ at all doping including the higher-energy dispersive band, except in the vicinity of $\nu = \pm2$, thus validating the estimation of $\Delta T$ from Mott fitting[25]. The $S \propto T$ behavior is expected in a degenerate weakly or non-interacting metal within the semiclassical framework, and has been verified for monolayer graphene[17] as well as tBLG at slightly larger $\theta$ ($2° \lesssim \theta \lesssim 5°$)[28]. Close to $\nu = \pm2$, however, we find an unexpected increase in $S$ when temperature is decreased below $\sim 40$ K, in contrast to the expectation of $S \approx 0$ (inset of Fig. 3d) from SMR and approaches $S \approx \pm2\mu$V/K for $\nu = \pm2$ respectively, at low $T$ (Figs. 3d and 4b). This is remarkable because, (1) at low $T$, the observed sign of $V_{2\omega}$ can not be assigned to the electron(hole)-like bands any more, and (2) the excess $S$ persists to a temperature scale ($\sim 40$ K) that is much higher than the superconducting transition ($T_c \sim 1.7$ K) in tBLG at $\theta = \theta_m$ or the temperature scale for correlated insulator ($\lesssim 4$ K)[4,6,11], suggesting a very distinct nature of the ground state. The absolute magnitude of the excess thermopower at $\nu = \pm2$ decreases with increasing $\theta$, as illustrated for a device with $\theta = 1.7°$ in Fig. 3c, and becomes undetectable for $\theta \gtrsim 2°$.

## Discussion

Although the Mott formula has been verified in a range of graphene-based devices[17,25], it can be violated in the hydrodynamic regime[29] and due to phonon drag in cross-plane thermoelectric transport in tBLG at $\theta > 6°$[20]. While the hydrodynamic regime is expected to appear at higher temperatures (>100 K), we eliminate the possibility of phonon drag from the observation of $S \propto T$ (away from $\nu = \pm2$, Fig. 3d). Furthermore, as shown in Fig. 4a, the occurrence of excess $S$, normalized as $(S - S_{Mott})/S_{max}$, where $S_{max}$ is the maximum value of $S$ at a given $T$, is concentrated in the low $T$ dome-like regions around $\nu = \pm2$ in the $T - (\nu, n)$ phase diagram. Since neither adiabatic (static) nor dynamical phonon effects can violate Mott formula[30,31], the enhancement of thermopower beyond the SMR limit suggests the possibility of a many body ground state similar to NFL phases in correlated oxides[32] and heavy Fermions[33]. A near-ubiquitous feature of the NFL regime in itinerant Fermionic systems, ranging from

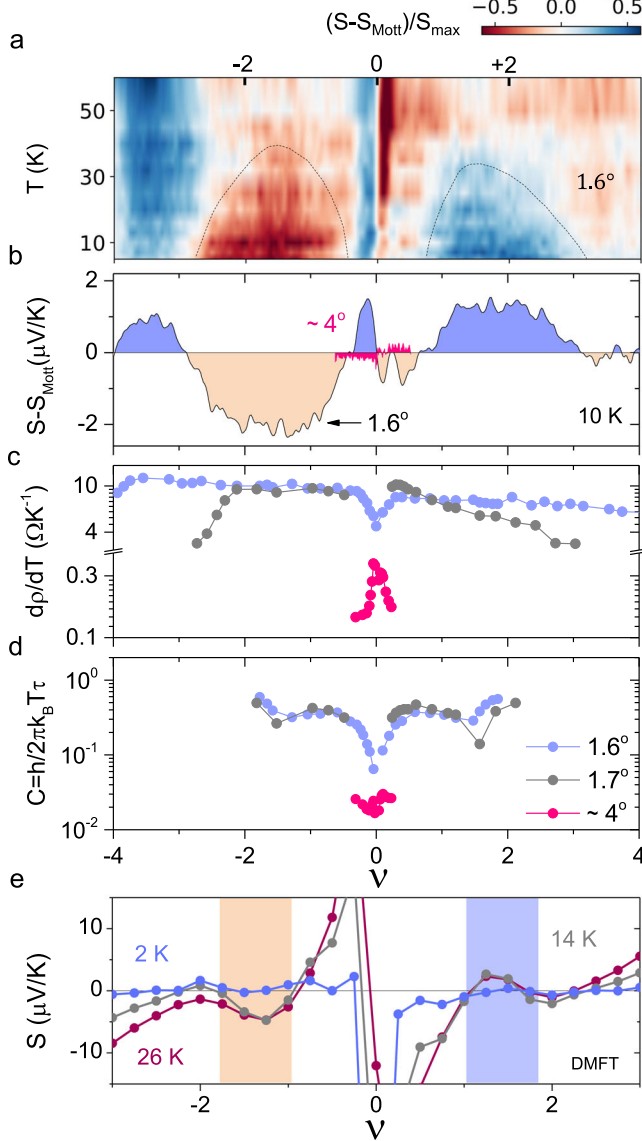

**Fig. 4 Breakdown of semiclassical Mott relation and scattering rate.**
**a** Surface plot of $(S - S_{Mott})/S_{max}$ as a function of $T$ and $\nu$ for $\theta \approx 1.6°$.
**b** $(S - S_{Mott})$ at 10 K for $\theta \approx 1.6°$ and $\theta \approx 4°$. **c** $d\rho/dT$ extracted in the $T$-linear regime at different $\nu$ for the three twist angles. **d** The estimated dimensionless pre-factor $C$ of the scattering rate $\Gamma = Ck_BT/\hbar$ as a function of $\nu$. **e** Seebeck coefficient $S$ computed in DMFT with $U = 38$ meV as a function of filling $\nu$ for the four lowest bands at three temperatures $T = 2$ K, 14 K and 26 K, respectively.

cuprates[34], ruthanates[35], pnictides[36] to heavy Fermions[33], is the 'strange metal' phase, characterized by the absence of well defined quasiparticles and linear $T$ dependence of $\rho$. Theoretical work also suggests possibilities of excess entropy, analogous to Bekenstein-Hawking entropy in charged black holes, in this regime, that remains finite down to vanishingly small $T$[37].

To check the mutuality between the excess thermopower and the strange metallic behaviour, we compare the $\nu$-dependence of excess $S$ at $T = 10$ K (Fig. 4b), and the scattering rate obtained from the slope $d\rho/dT$ in the $T$-dependence of $\rho$ (Fig. 4c). For reference, we also present the results from another device at $\theta \approx 4°$, where we find no violation of SMR over the experimental range of $n$. In the NFL state, the incoherent scattering rate is $\tau^{-1} = Ck_BT/\hbar$, where the dimensionless coefficient $C$ is of the order of unity for Planckian

dissipation. In Fig. 4c we plot the $\nu$-dependence of $d\rho/dT$ and $C$ (Fig. 4d), where $C$ is computed from $d\rho/dT$ assuming Drude-like resistivity in accordance to Refs. [8,9] (See Supplementary Note 7). Away from the CNP, both 1.6° and 1.7° devices show $d\rho/dT \approx 10$ $\Omega$/K near $\nu \approx \pm 2$, which is nearly two orders of magnitude larger than $d\rho/dT \approx 0.2$–0.3 $\Omega$/K for the tBLG device at $\theta \approx 4°$, implying that the individual layers are essentially decoupled in the latter[8,10]. Intriguingly, for tBLG at $\theta = 1.6°$ and 1.7°, we find $C$ to approach the order of unity in the vicinity of $\nu \rightarrow \pm 2$, raising the possibility of a common physical origin for the violation of SMR. Notably, the excess thermopower was found largely unaffected in the in-plane magnetic field (See Supplementary Fig. 20), and thus unlikely to arise from an underlying spin/magnetic texture[5]. Theoretically a dynamical mean field theory (DMFT)[3,38,39] calculation shows qualitative agreement in the density dependence of excess thermopower at $\nu = \pm 2$ but fails to capture its finite magnitude at low temperature (Fig. 4e). This is because the particular single site DMFT framework used in our calculation would invariably lead to the FL phase as $T \rightarrow 0$, even though some excess thermopower can be observed in the intermediate temperature range (See Supplementary Note 8). Particle-hole asymmetry due to $\Omega/T$ ($\Omega$ is the energy of an excitation counted from Fermi level) scaling in certain non-Fermi liquid Planckian metals, on the other hand, may not only cause a logarithmically divergent S at low T (Fig. 3d, inset), but also a sign reversal in S for the electron- and hole-type bands[40,41].

In summary, we have measured the electrical resistivity and thermopower in twisted bilayer graphene over a broad range of low-twist angles. At larger $\theta$ ($\sim 1.6°$ − 1.7°), our experimental results show concurrent $T$-linear resistivity at Planckian dissipation scales and emergent excess thermopower below $T \lesssim 40$ K near $\nu = \pm 2$ signifying the breakdown of the semiclassical Mott relation. The thermopower near $\nu = \pm 2$ approaches a finite magnitude ($\approx 2 \mu$V/K at 1.6°) at low T providing a new facet to the strongly correlated 'strange metal' phase in tBLG. Our experimental results point to a truly non-Fermi liquid (NFL) metallic state in tBLG at low twist angle that carry strong similarities to those observed in cuprates or heavy-Fermion materials with low coherence temperatures.

## Methods

**Device fabrication**. All devices in this work were fabricated using a layer-by-layer mechanical transfer method[20]. Monolayer graphene and hexagonal boron nitride (hBN) were exfoliated on $SiO_2$/Si wafers and graphene flakes were identified using optical microscopy and Raman spectroscopy. For $\theta \approx 1.6°$, the edges of the graphene flakes were aligned under an optical microscope and encapsulated within two hBN layers. Other tBLG devices were fabricated using tear and stack method[42]. Electron beam lithography was used to define Cr/Au top gate for tuning the number density in the tBLG region. Finally, the electrical contacts were patterned by electron-beam lithography and reactive ion etching followed by metal deposition (5 nm Cr/50 nm Au) using thermal evaporation technique.

Electrical transport measurements were performed in a four-terminal geometry with typical ac current excitations of 10–100 nA using a standard low-frequency lock-in amplifier at 226 Hz, in a dilution refrigerator and a 1.5-K cryostat. For thermoelectric measurements, local Joule heating was employed to create a $\Delta T$ across the tBLG channel. A range of sinusoidal currents (2–5 $\mu$A) at excitation frequency $\omega = 17$ Hz were used for Joule heating and the resulting 2nd harmonic thermal voltage ($V_{2\omega}$) was recorded using a lock-in amplifier. Thermoelectric measurements were conducted in a 1.5-K cryostat/20 mK dilution refrigerator with magnetic field.

**Tight binding calculation of DOS**. The rigid bilayer structures were generated using the Twister code[43]. The structures were subsequently relaxed in LAMMPS[44,45] using REBO[46] as the intralayer potential and DRIP[47] as the interlayer potential. These relaxed structures were used for performing all the calculations.

The electronic band structures were calculated by approximating the tight binding transfer integrals under the Slater Koster formalism[48]. A more detailed discussion of the calculations is available in the Supplementary Note 8.

## Data availability
Source data are available for this paper. All other data that support the plots within this paper and other findings of this study are available from the corresponding author upon reasonable request. Source data are provided with this paper.

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

## Acknowledgements
The authors thank Nano mission, DST for the financial support. M.J. and S.M. thank the computational facilities in SERC. K.W. and T.T. acknowledge support from the Elemental Strategy Initiative conducted by the MEXT, Japan, Grant Number JPMXP0112101001, JSPS KAKENHI Grant Numbers JP20H00354 and the CREST(JPMJCR15F3), JST. U.C. acknowledges funding from IISc and SERB (ECR/2017/001566), and H.R.K. from SERB(SB/DF/005/2017). S.B. acknowledges funding from IISc and SERB (ECR/2018/001742).

## Author contributions
B.G., P.S.M. and M.G. contributed equally to this work. B.G., M.G. and SA.B. fabricated the devices with help from R.S. The transport measurements were performed by B.G., P.S.M. and M.G. with help from A.J. The results were analysed by B.G., P.S.M. and M.G. S.M., M.J., H.R.K. and S.B. provided the presented theory calculations. Hexagonal boron nitride crystals were grown by K.W. and T.T. A.G. and U.C. contributed in the data interpretation, and theoretical understanding of the manuscript. B.G., P.S.M. and A.G. wrote the manuscript with inputs from all authors.

## Competing interests
The authors declare no competing interests.
