## [Peer Review File · Nature Communications]

REVIEWER COMMENTS

Reviewer #2 (Remarks to the Author):

The paper investigates twisted bilayer graphene using measurements of current driven by the voltage and temperature gradient. It reports interesting large values of thermoelectric response close to filling factor $\nu=2$, uncovering a new facet of the correlated state there. The subject of manuscript is timely and the main result interesting. I recommend the paper for publication after revision.

I have few questions and suggestions:

1. the discussion around Eq 1 needs to be reconsidered. To my knowledge, the Mott formula actually assumes quasiparticle picture (description of the response using the Boltzmann formalism in relaxation time approximation) and a Sommerfeld expansion, but not much more. I think momentum dependence of the scattering rate could be incorporated into transport function and hence isotropy assumption is not essential. Authors might also reconsider substituting term "energy-dependent" with "Fermi energy dependent", because in Mott formula one has in mind the derivative with the Fermi energy. If indeed isotropy and momentum-dependence are needed, it might be good to add a reference.
2. Where the authors write "exhibit additional mimaxima in R near integer values of ν ", reader is confused because looking at Fig 1 a, these maxima are not really at integer values. In the next sentence then they discuss also shift, but perhaps it would be better to already split the first sentence saying first that they exhibit additional maxima and then later where they are.
3. I do not understand how $T < T_F$ can hold at $\nu=4,0$, where one has a band insulator. [Sentence "We also note that $T \ll T_F$ is maintained..."]. It is also difficult to understand how ΔT that is extracted assuming the Mott relation and then applying it to extract thermopower also when the Mott relation is according to authors violated. If the procedure is robust, the authors should detail why, if it is not, the authors should clarify what is robust and what not.
4. typo : thermovolatge
5. I find two recent preprints on thermoelectric effects in non-Fermi liquids are related to findings of the paper: arXiv:2106.05959 and in particular arXiv:2102.13224 . Namely, given the fact that elastic

scattering clearly dominates the resistivity in the low temperature regime, a reader may wonder how one can have excess thermopower in this regime, and this is addressed in those works.

Reviewer #4 (Remarks to the Author):

This manuscript written by Ghawri presented thermopower measurement of twisted bilayer graphene to explore the violation of Mott formula in integer filling below $\nu = 4$. First, let me put all the reviewer's comments in the previous round of review aside, and provide my general comment after reading the manuscript. As the author stated, the comparison of thermopower between experiment and Mott formula could, in principle, provide the information about the deviation of experimental data from the Mott formula. And that could be due to the different mechanisms in the scattering process. In that regard, the objective of this study sounds interesting since it may allow researchers to access the information on the unique band structure or interaction-related phenomena in twisted bilayer graphene. However, the manuscript does not seem to provide robust and clear evidence to support their conclusion. I am afraid that to draw their conclusion, the carefully controlled measurement to allow one to perform quantitative comparison is necessary; however, as I comment in detail on the later paragraph, unfortunately, their measurement does not seem convincing enough to reach that conclusion.

Following are my concerns which I would like to ask the author to clarify.

Comment 1.

I am afraid that their measurement device geometry may not have the homogeneous temperature gradient that they assumed. The heater part of the graphene region is connected to the bilayer graphene region and these are both encapsulated by the same h-BN insulator. In addition, there is a large metal top gate above BN. The graphene and BN are both reasonably good thermal conductors even at low temperatures. Thus, the real temperature gradient may not be the same as what the author assumed. The four terminal resistance is measured under the homogeneous flow of current; thus measuring the average of the channel region. However, the temperature gradient could be more inhomogeneous and it could be measuring very local regions in the worst case. My concern is the real temperature gradient may be localized to one of the interfaces not homogeneous that the author assumed. I admit that overall shape change such that where the peak or dip in the thermopower can be still reasonably discussed, however it may be difficult to achieve quantitative comparison. As ref. [17] or other paper showed, thermopower measurement occasionally requires careful isolation between heater and device to allow homogeneous temperature gradient and careful elimination of substrate-induced effect.

I am concerned about this since their supplement figure S8 showed that D3 sometimes showed resistance peaks around $\nu = 2$. So this device may have some states around here that may become

sensitive with thermopower due to the possible locality of the measurement but was not visible with resistance measurement since it is global measurement.

I found that one of the reviewers was concerned about a similar kind of temperature gradient issue (comment 2 of reviewer 2), but I do not think the author sufficiently answered that question. The problem is that the heater and detector region was connected by the region where carrier density is tuned by a top gate that is shared with twisted bilayer graphene. The heat transfer through this region is not guaranteed to be ineffective to their measurement.

To substantiate their conclusion, I think authors really need to measure the temperature gradient and temperature rise in their device to check that their resistance and thermopower originated from the same region with the information of actual temperature rise.

Comment 2.

Their thermopower is due to the seebeck effect such that it changes the sign of thermopower across the charge neutrality and it is 0 at the charge neutrality. Similar but opposite sign change will take place for the case of van Hove singularity. Then, it is not clear what kind of mechanism can create the sign of the deviation from mott formula presented in Fig. 4b. They discussed the positive dome-shape peak in the electron doped side and dip in hole side. So the excess signal tends to be mono polar. Then there is no explanation why his excess signal have to be positive or negative according to their interpretation. I also wondered why it does not have any sign change around $\nu = +2$ and -2 . Could the author provide an explanation for this. The phonon drag they mentioned, in principle, can generate a single peak or dip-like change in thermopower(if I understand correctly); however I still want to see the discussion why it has to be a peak in the electron side and dip in hole-side.

Comment 3.

Although this point was already suggested by other reviewers, I am also afraid of the quality of the measured sample. The resistance data in Fig. 1 tend to be broad and the twist angle dependent property is not so obvious to me. In addition, the device geometry is different for D3. Conclusion obtained from twist angle dependence is rather questionable. It is true that obtaining high quality samples in twisted bilayer graphene is still a challenging subject, but just by looking at the data, I do not feel that their device and measurement reach the level to obtain a precise comparison with the Mott formula.

Comment 4.

Since in any case, the thermopower is still related to the carrier scattering, one would expect some correlation between the figures between Fig. 3d and similar plot with Resistance as a y-axis. Does the author obtain any correlation between such figures? If it does not, could the author explain the reason.

Comment 5.

In twisted bilayer graphene samples, some of the states become more visible in the magnetic field. For the thermopower measurement, the Seebeck effect does not change its sign under the magnetic field reversal and probably phonon drag thermopower does not as well. Instead, if there is Nernst contribution in their device, it will experience sign change under magnetic field reversal. Therefore, I think the magnetic field dependence of thermopower and comparing it with magnetic field dependence of resistance may provide some extra information to understand the nature of the device. Would it be possible to present such data?

Reviewer #2

The paper investigates twisted bilayer graphene using measurements of current driven by the voltage and temperature gradient. It reports interesting large values of thermoelectric response close to filling factor $\nu=2$, uncovering a new facet of the correlated state there. The subject of manuscript is timely and the main result interesting. I recommend the paper for publication after revision.

I have few questions and suggestions:

1. the discussion around Eq 1 needs to be reconsidered. To my knowledge, the Mott formula actually assumes quasiparticle picture (description of the response using the Boltzmann formalism in relaxation time approximation) and a Sommerfeld expansion, but not much more. I think momentum dependence of the scattering rate could be incorporated into transport function and hence isotropy assumption is not essential. Authors might also reconsider substituting term "energy-dependent" with "Fermi energy dependent", because in Mott formula one has in mind the derivative with the Fermi energy. If indeed isotropy and momentum-dependence are needed, it might be good to add a reference.

Response: We agree with the referee. The derivation of Mott formula only requires the existence of well-defined quasiparticle, semiclassical approximation for their dynamics and elastic scattering, which leads to relaxation time approximation. The relaxation time could be momentum dependent, only dependence on energy is not required. We have modified this statement in the revised manuscript.

Changes in manuscript:

"Eq. 1 is valid under the assumption that scattering is elastic and isotropic close to the Fermi surface i.e., the transport lifetime only depends on the energy of the charge carriers. Remarkably, this simple assumption of isotropic scattering remains valid in a wide variety of systems .."

To

"Eq.1 is valid for a quasiparticle description of transport using semiclassical Boltzmann equation under the assumption that scattering is elastic close to Fermi surface. Remarkably, this simple assumption of elastic quasiparticle scattering remain valid in a wide variety of systems .."

2. Where the authors write "exhibit additional mimaxima in R near integer values of ν ", reader is confused because looking at Fig 1 a, these maxima are not really at integer values. In the next sentence then they discuss also shift, but perhaps it would be better to already split the first sentence saying first that they exhibit additional maxima and then later where they are.

Response: We thank the reviewer for the suggestion. We note that the device with $\theta \approx 1.01^\circ$ exhibits maxima in resistance exactly at integer filling factors, whereas resistance peaks for $\theta \approx 1.24^\circ$ are shifted slightly away from the integer fillings at base $T \sim 100$ mK (Fig. S4). In addition, we observe a substantial shift $|\Delta \nu| \sim 0.25$ of resistance peaks near $\nu = +1$ and $+3$ from 7 K to 35 K, suggesting the possibility of isospin-polarization in the system (SI, section III). We have clarified this in the revised manuscript.

Changes in manuscript: The following sentence has been included in the manuscript:

“In addition, near θ_m , the device with $\theta \approx 1.01^\circ$ exhibit additional maxima in R at integer values of ν , whereas for $\theta \approx 1.24^\circ$ resistance peaks are shifted slightly away from integer fillings”.

3. I do not understand how $T < T_F$ can hold at $\nu=4,0$, where one has a band insulator. [Sentence "We also note that $T \ll T_F$ is maintained..."]. It is also difficult to understand how ΔT that is extracted assuming the Mott relation and then applying it to extract thermopower also when the Mott relation is according to authors violated. If the procedure is robust, the authors should detail why, if it is not, the authors should clarify what is robust and what not.

Response:

We wish to emphasize following points:

1. We agree with the reviewer that at CNP and $\nu = \pm 4$, T_F can be substantially small and violate the degenerate limit $T \ll T_F$. However, we perform the fitting at the local maxima/minima of $V_{2\omega}$ which is slightly away from the CNP ($n \approx 0.6 \times 10^{12} \text{ cm}^{-2}$), where we find $T_F \approx 240 \text{ K}$ from the band structure calculation (Fig. R1). This satisfies the degenerate limit $T \ll T_F$ required for the validity of the Mott relation. Similarly, near the local maxima close to $|\nu| = 4$ ($n \approx 7.6 \times 10^{12} \text{ cm}^{-2}$), the value of $T_F \approx 250 \text{ K}$ and satisfies the degenerate limit. We have clarified this important point in the revised manuscript and supplementary information.

Changes in the manuscript:

For fitting Eq. 2, we also note that $T \ll T_F$ is maintained throughout almost the entire temperature and gate voltage range shown in Fig. 3b, except very close to CNP ($\nu = 0$) and $\nu = \pm 4$ (see SI, section V).

Fig. R1 Doping dependence of T_F (left axis) calculated using tight-binding model plotted with the measured $V_{2\omega}$ and calculated thermopower using Mott formula (right axis). The Fermi temperature (T_F) is obtained from the band structure calculation by converting the energy scale to temperature.

2. This employed method of obtaining ΔT from Mott fitting is robust and frequently used in graphene-based devices (see I. J. Vera-Marun et al. *Nat Commun* **7**, 11525 (2016), Jayaraman et al. *Nano Lett.* **21**, 1221 (2021) and Sierra et al. *Nature Nanotech* **13**, 107–111 (2018)). Using ΔT as the single fitting parameter, we obtain excellent agreement between the measured $V_{2\omega}$ and that calculated using Mott formula near CNP, $\nu \sim \pm 4$, and also in the higher energy dispersive band ($\nu > \pm 4$) simultaneously (See SI, Fig. S16). This suggests that the temperature gradient (ΔT) in the tBLG channel is independent of the doping and only depends on the heating current at the SLG region. We also note that the observed agreement between measured thermopower and Mott formula is expected in absence of correlation effects, which is the case near CNP and in higher dispersive bands, thereby justifying our method of obtaining ΔT . As evident from the Fig. S17, the obtained Seebeck coefficient ($= V_{2\omega}/\Delta T$) shows T-linear dependence in the dispersive band ($\nu = -5, -6, -7$) as expected from the T-linear dependence from Mott relation, validating the employed method of estimating ΔT from the Mott fitting in our experiments.

To further justify our method, we show thermopower data from a different twisted bilayer graphene sample, where we have used resistance thermometry to calculate the ΔT independently. We find excellent agreement between the ΔT obtained using the resistance thermometry and by fitting the Mott formula to the experimental data as shown in Fig. R4(a) and (b).

Changes in manuscript:

The following sentences have been added to the supplementary information (section V. C):

1. “This employed method of obtaining ΔT from Mott fitting is robust and frequently used in graphene-based devices (I. J. Vera-Marun et al. *Nat Commun* **7**, 11525 (2016), Jayaraman et al. *Nano Lett.* **21**, 1221 (2021) and Sierra et al. *Nature Nanotech* **13**, 107–111 (2018)).”
2. “This suggests that the temperature gradient (ΔT) in the tBLG channel is independent of the doping and only depends on the heating current at the SLG region. We also note that the observed agreement between measured thermopower and Mott formula is expected in absence of correlation effects, which is the case near CNP and in higher dispersive bands, thereby justifying our method of obtaining ΔT .”

4. typo : thermovolatge

Response:

We thank the referee for pointing this out. We have corrected this in the revised manuscript.

5. I find two recent preprints on thermoelectric effects in non-Fermi liquids are related to findings of the paper: arXiv:2106.05959 and in particular arXiv:2102.13224 . Namely, given the fact that elastic scattering clearly dominates the resistivity in the low temperature regime, a reader may wonder how one can have excess thermopower in this regime, and this is addressed in those works.

Response: We thank the referee for suggesting these important references and have included in the revised manuscript.

Changes in manuscript:

Particle-hole asymmetry due to Ω/T (Ω is the energy of an excitation counted from Fermi level) scaling in certain non-Fermi liquid Planckian metals, on the other hand, may not only cause a logarithmically divergent S at low T (Fig. 3d, inset), but also a sign reversal in S for the electron- and hole-type bands [REF: arXiv:2106.05959 and arXiv:2102.13224].

Reviewer #4

This manuscript written by Ghawri presented thermopower measurement of twisted bilayer graphene to explore the violation of Mott formula in integer filling below $\nu = 4$. First, let me put all the reviewer's comments in the previous round of review aside, and provide my general comment after reading the manuscript. As the author stated, the comparison of thermopower between experiment and Mott formula could, in principle, provide the information about the deviation of experimental data from the Mott formula. And that could be due to the different mechanisms in the scattering process. In that regard, the objective of this study sounds interesting since it may allow researchers to access the information on the unique band structure or interaction-related phenomena in twisted bilayer graphene. However, the manuscript does not seem to provide robust and clear evidence to support their conclusion. I am afraid that to draw their conclusion, the carefully controlled measurement to allow one to perform quantitative comparison is necessary; however, as I comment in detail on the later paragraph, unfortunately, their measurement does not seem convincing enough to reach that conclusion.

Following are my concerns which I would like to ask the author to clarify.

We thank the reviewer for reviewing the manuscript, and for giving valuable comments. The responses to the comments are as follows

Comment 1.

I am afraid that their measurement device geometry may not have the homogeneous temperature gradient that they assumed. The heater part of the graphene region is connected to the bilayer graphene region and these are both encapsulated by the same h-BN insulator. In addition, there is a large metal top gate above BN. The graphene and BN are both reasonably good thermal conductors even at low temperatures. Thus, the real temperature gradient may not be the same as what the author assumed. The four terminal resistance is measured under the homogeneous flow of current; thus measuring the average of the channel region. However, the temperature gradient could be more inhomogeneous and it could be measuring very local regions in the worst case. My concern is the real temperature gradient may be localized to one of the interfaces not homogeneous that the author assumed. I admit that overall shape change such that where the peak or dip in the thermopower can be still reasonably discussed, however it may be difficult to achieve quantitative comparison. As ref. [17] or other paper showed, thermopower measurement occasionally requires careful isolation between heater and device to allow homogeneous temperature gradient and careful elimination of substrate-induced effect.

Response: We thank the referee for this important question. We wish to clarify following points:

1. While it is true that some studies do employ external heater, we wish to emphasise that the employed method of generating ΔT by heating one side of the device and measuring the thermal voltage across the channel has also been frequently used in 2D systems such as GaAs-AlGaAs (Scheibner et al. PRL 95, 176602 (2005), Goswami et al. PRL 103, 026602 (2009)) and in graphene-based devices (see Jayaraman et al. Nano Lett. 21, 1221 (2021)). We also note that a recent study by Philip Kim's group (Waissman et al. *Nat. Nanotechnol.* (2021)) employed similar method to create ΔT across a hBN-Gr-hBN heterostructure device and reported a homogeneous ΔT in the device. Additionally, at ultra-low temperature, thermal conductivity of Si substrate is quite low to generate sufficient ΔT , and thus connected heater seems to be better way to achieve a reasonable ΔT .
2. We note that in our experiments a single fitting parameter (ΔT) is used for fitting the Mott formula in the entire density range, which gives excellent agreement between the measured $V_{2\omega}$ and that calculated using Mott formula near CNP, $\nu \sim \pm 4$, and also in the higher energy dispersive band ($\nu > \pm 4$) simultaneously. In case of an inhomogeneous ΔT , different regions will contribute differently to measured thermopower and it will not be possible to fit the experimental data with one parameter, which suggests a rather homogeneous ΔT in the device. To further substantiate our argument, we have shown thermopower data at 82 K (Fig. R2) from another twisted graphene device (twist angle= 0° , Bernal stacking), where measured thermovoltage follows the Mott relation reasonably well and hence justifies the device geometry.

Fig. R2 Gate voltage dependence of measured $V_{2\omega}$ for a twisted graphene device (twist angle= 0° , Bernal stacking) compared with that calculated using Mott formula.

Changes in manuscript:

The following sentences have been added to the supplementary information (section IV):

“We wish to emphasize that the employed method of generating ΔT by heating one side of the device and measuring the thermal voltage across the channel has also been frequently used in 2D systems such as GaAs-AlGaAs (Scheibner et al. PRL 95, 176602 (2005), Goswami et al. PRL 103, 026602 (2009)) and in graphene-based devices (Jayaraman et al. Nano Lett. 21, 1221 (2021), Waissman et al. *Nat. Nanotechnol.* (2021)). It is to be noted that at ultra-low temperature, thermal conductivity of Si substrate is quite low to generate sufficient ΔT , and thus connected heater seems to be better way to achieve a reasonable ΔT .”

I am concerned about this since their supplement figure S8 showed that D3 sometimes showed resistance peaks around $\nu = 2$. So this device may have some states around here that may become sensitive with thermopower due to the possible locality of the measurement but was not visible with resistance measurement since it is global measurement. I found that one of the reviewers was concerned about a similar kind of temperature gradient issue (comment 2 of reviewer 2), but I do not think the author sufficiently answered that question. The problem is that the heater and detector region was connected by the region where carrier density is tuned by a top gate that is shared with twisted bilayer graphene. The heat transfer through this region is not guaranteed to be ineffective to their measurement. To substantiate their conclusion, I think authors really need to measure the temperature gradient and temperature rise in their device to check that their resistance and thermopower originated from the same region with the information of actual temperature rise.

The Referee has made an important observation. We clarify the peak in thermopower and experimental arrangement below:

1. The peak in resistance around $\nu \sim \pm 2$ for D3 in Fig. S8 can be attributed to van Hove singularities in the band structure of tBLG, but it cannot give rise to a ‘peak’ or ‘trough’ in thermopower, it must show a zero crossing at these filling, as observed in 1.24⁰ device. Hence the origin of the excess thermopower cannot be connected to a local thermal gradient.
2. We have observed the opposite sign of the excess thermopower (ΔS) on eI- and h-sides in multiple contact configurations for the 1.6⁰ device (Fig. R3a), which rules out the possible local variation of thermopower measurements as suggested by the referee. This data is presented in the supplementary information (Fig. S23) for clarification. Additionally, we have shown the thermoelectric measurements in a different device fabricated in hall-bar geometry with $\theta \approx 1.7^\circ$. We find that at low temperature, the thermopower measurements show an excess S with opposite sign of ΔS on the electron and hole sides as shown in Fig. R3b below.

Fig. R3 (a) Optical micrograph of the device D3 with the contact configuration and measured $V_{2\omega}$ for three different heating configurations. (b) Doping dependence of Seebeck coefficient for device D4 compared to that obtained using Mott formula at 5 K. Contacts used are marked in the optical micrograph.

3. We wish to clarify that the heater region (SLG) and the tbLG channel do not share the top gate. Heater region is kept at a fixed density while the top gate tunes the number density only in the twisted region.
4. It is important to note that we measure/estimate the real temperature gradient. There is no question of assumption. To justify our method of obtaining ΔT , we show thermopower data from a different twisted bilayer graphene sample ($\theta \sim 2^\circ$), where we have used resistance thermometry to calculate the ΔT independently. We find excellent agreement between the measured $V_{2\omega}$ and that calculated using Mott formula in the entire density range as shown in Fig. R4a. In Fig. R4b, we plot ΔT obtained using the resistance thermometry and by fitting the Mott formula to the experimental data. Excellent agreement between the two justifies our method of obtaining ΔT . We note that the resistance thermometry is not particularly effective at low temperature and hence we have used Mott formula to obtain the ΔT in the current manuscript. Moreover, as the Referee notes her/himself, the exact value of the temperature gradient is not of crucial importance to any of the conclusions in the paper

Fig. R4 (a) The doping dependence of experimentally measured S (connected coloured circles) compared to S_{Mott} (black solid line). (b) Comparison of ΔT obtained from resistance thermometry and that extracted using Mott formula at various T .

Changes in manuscript:

1. The following sentence has been added to the supplementary information (section III. D):

“We wish to emphasise that the peaks in resistance near $\nu \sim \pm 2$ cannot give rise to a ‘peak’ or ‘trough’ in thermopower, it must show a zero crossing at these filling, as observed in 1.24° device. Hence the origin of the excess thermopower cannot be connected to the appearance of these peaks.”

2. Data shown in Fig. R4 has been presented in Fig. S18 in supplementary information

Comment 2.

Their thermopower is due to the seebeck effect such that it changes the sign of thermopower across the charge neutrality and it is 0 at the charge neutrality. Similar but opposite sign change will take place for the case of van Hove singularity. Then, it is not clear what kind of mechanism can create the sign of the deviation from mott formula presented in Fig. 4b. They discussed the positive dome-shape peak in the electron doped side and dip in hole side. So the excess signal tends to be mono polar. Then there is no explanation why his excess signal have to be positive or negative according to their interpretation. I also wondered why it does not have any sign change around $\nu = +2$ and -2 . Could the author provide an explanation for this. The phonon drag they mentioned, in principle, can generate a single peak or dip-like change in thermopower (if I understand correctly); however I still want to see the discussion why it has to be a peak in the electron side and dip in hole-side.

Response: We thank the referee for raising this important point. An explanation of the non-standard sign of the thermopower (and excess thermopower) as well as the absence of the sign change at $\nu \sim \pm 2$ was given based on the DMFT picture, at least for the intermediate temperatures and is discussed below as well as in detail in the SI (section VII.). Referee’s expectation of sign changes

around $\nu \sim \pm 2$ is based on a non-interacting picture. There is no reason for such expectation to be valid for a strongly correlated metallic state and that is the main message of the experimental results.

For a theoretical understanding, we explored the impact of electron interaction and vHS within a dynamical mean field theory (DMFT). Considering the four lowest bands near the CNP and a Hubbard interaction $U = 0.2W$, we find that the low-energy vHSs enhance the effect of interaction for fillings $|\nu| \simeq 1 - 2$, with a strongly particle-hole asymmetric electronic self-energy at low energies and a low coherence temperature (T_{coh}) scale, only below which the system behaves as FL (SI, section VII for details). The strong particle-hole asymmetric self-energy effects near the vHSs lead to large deviations and different sign of S (Fig. 4e, main text) from the non-interacting thermopower (Fig.S30, section VII, SI) $|\nu| \simeq 1 - 2$, as explained in detail in the SI (section VII.). However, within the particular single-site DMFT framework used in our calculations, one inevitably gets a FL for $T \rightarrow 0$. Hence the DMFT results cannot capture the finite $S(\nu = \pm 2)$, the non-standard excess thermopower and the absence of sign change at very low-temperature, as seen in experiment. Thus, the positive dome-shape peak in the electron doped side and dip in hole side, the mono polar excess thermopower and the absence of sign change around $\nu = +2$ and -2 are mainly experimental observations and point towards an unusual correlated metallic state, which defy a rationalization based on non-interacting band-structure-based picture.

We agree with the referee that phonon drag can give a single peak or dip-like change in thermopower, however there is no reason to believe that phonon drag will appear only in the vicinity of $\nu \sim \pm 2$ and nowhere else. Hence, we eliminate the possibility of phonon drag from the observation of $S \propto T$ (away from $\nu = \pm 2$, Fig. 3d).

Comment 3.

Although this point was already suggested by other reviewers, I am also afraid of the quality of the measured sample. The resistance data in Fig. 1 tend to be broad and the twist angle dependent property is not so obvious to me. In addition, the device geometry is different for D3. Conclusion obtained from twist angle dependence is rather questionable. It is true that obtaining high quality samples in twisted bilayer graphene is still a challenging subject, but just by looking at the data, I do not feel that their device and measurement reach the level to obtain a precise comparison with the Mott formula.

Response:

We appreciate the Referee's concern on the quality of the devices, which we have addressed quantitatively below. However, we wish to emphasize that the 'quality' of the device does not prohibit a comparison with Mott formula. Mott formula is understood to be robust against impurity scattering in rather broad class of diffusive semi-classical systems, and has been established even in early graphene devices, with carrier mobility much lower than ours (PRL 102, 096807 (2009)) Among other sources, disorder in our devices may also arise from spatial fluctuations in twist angle, as generally observed in this type of devices, which act as an additional source of scattering (arXiv:1908.02753). However, the comparison with the Mott formula seems justified. We wish to clarify following points:

1. We wish to mention that R-T in fig. 1a of the manuscript is shown in log-linear (y-x) scale which makes the resistance data look broader. To elucidate this point, we have shown R-T data both in log-linear as well as in linear-linear scale. (Fig. R5 below). Additionally, the temperature range (3-70 K) of our interest leads to significant thermal broadening.

Fig. R5 Temperature-dependent resistance R as a function of band filling ν for four different devices with twist angle 1.01° , 1.24° , 1.6° and 1.7° in (a) log-linear (y - x) scale (b) in linear-linear (y - x) scale

2. We wish to emphasize following points regarding the device quality,

- (a) Fig. R6 shows the two-terminal resistance measurement in device D1 between various neighbouring pairs of contacts. The device exhibits consistent twist angle ($1.01^\circ \pm 0.02^\circ$) between all contacts, as evidenced by very good alignment of resistive states at integer fillings. The observed twist angle homogeneity is consistent with previous studies on high quality tBLG samples (Saito, Y. *et al. Nat. Phys.* **16**, 926–930 (2020), Arora, H.S., *et al. Nature* **583**, 379–384 (2020)).

Fig. R6 (a) Optical micrograph of the device D1. The scale bar represents a length of 2 μm. (b) Resistance as a function of number density measured between different pair of contacts at 4.8 K.

(b) In Fig. R7, we have shown the resistance data for different devices taken from previous studies. Fig. R7a, b and c show the data for devices close to magic angle, which show similar full width half maxima (FWHM) of the resistance peak at the CNP. For example, our devices with 1.01° and 1.24° show a FWHM of 0.14×10^{12} and 0.12×10^{12} cm⁻² respectively at the CNP, which is similar to resistance data (FWHM $\sim 0.1 \times 10^{12}$ cm⁻²) shown in Fig. R7a. Further, R7d shows resistance data from a device at 1.8°, which shows similar transfer characteristics as our higher angle devices (D3 and D4). This suggests that our device quality is similar to what is reported in literature before, and the results shown in this study are intrinsic to tBLG samples.

Fig. R7 (a), (b), (c) and (d) Electric transport data for different devices studied in references [1], [2], [3] and [4] respectively.

3. We note that although the device geometry is different for D3, overall measurement scheme remains the same as we create ΔT by heating the SLG channel outside the top gated (tBLG) region. Despite the different geometries, our experiments reveal that the two devices ($\approx 1.6^\circ$ and $\approx 1.7^\circ$) exhibit near-identical violation of Mott relation around $\nu \sim \pm 2$ at low T, which highlights the intrinsic characteristics of the tBLG devices presented in the manuscript

Referee: "Conclusion obtained from twist angle dependence is rather questionable"

We have performed thermopower measurements in devices over a broad range of low- twist angles. However, for magic angle devices, our experiments revealed that near integer filling of the moiré band, it is difficult to unambiguously distinguish the excess thermopower from the additional thermopower arising from the competing interactions at $\nu = \pm 2$ and a direct comparison with Mott formula becomes very difficult. Whereas in higher twist angle devices (away from magic angle), we can unambiguously detect the excess thermopower and a clear violation of Mott formula is observed. Furthermore, in the device near magic angle ($\theta \approx 1.24^\circ$) away from half-filling, we observe excess thermopower coinciding with T-linear resistivity at $\nu = 0.5$ where other competing interactions are relatively weak.

Comment 4.

Since in any case, the thermopower is still related to the carrier scattering, one would expect some correlation between the figures between Fig. 3d and similar plot with Resistance as a y-axis. Does the author obtain any correlation between such figures? If it does not, could the author explain the reason.

Response: We thank the referee for raising this important point. We wish to emphasize that both thermopower (other than at $\nu \sim \pm 2$) and resistance shows a linear-in-T behaviour below ~ 100 K (Fig. 1d, Fig. S9 and Fig. R8). However, we note that resistance remains T-linear at all densities, thermopower shows a clear departure from T-linearity around $\nu \sim \pm 2$ indicating violation of Mott formula. The correlation mentioned by the referee holds true only when resistance and thermopower are related by Mott formula, which indeed is true for our device away from half filling. However, at a more fundamental level, the Seebeck coefficient is a ratio of transport coefficients that involves states immediately below and above the Fermi level, in contrast with electrical transport, which is sensitive only to the properties of the Fermi surface. As a result, the Seebeck coefficient is controlled by the particle-hole asymmetry between occupied and unoccupied states around the Fermi level. This asymmetry can originate both from the dispersion of electronic excitations (band structure) and from the energy dependence of the scattering rate, which is not accounted by the Mott formula. Hence, it is not always possible to draw a correlation between T-dependence of Seebeck coefficient and resistance.

We attract the referee's attention to the recent theoretical work (REF: arXiv:2106.05959), noted by referee 2, where electron-hole asymmetry in the non-Fermi liquid Planckian regime, can lead to logarithmically divergent thermopower at low temperature. No correlation between the resistance and the thermopower is expected in this case. We have added a discussion on this in the modified manuscript.

Fig. R8 Temperature dependence of resistivity (ρ) at filling factors $\nu = \pm 2$. The inset shows T-dependence of $\rho - \rho_0$ at $\nu = \pm 2$ in logarithmic scale. Solid lines show T-linear and T^2 dependences, respectively.

Comment 5.

In twisted bilayer graphene samples, some of the states become more visible in the magnetic field. For the thermopower measurement, the Seebeck effect does not change its sign under the magnetic field reversal and probably phonon drag thermopower does not as well. Instead, if there is Nernst contribution in their device, it will experience sign change under magnetic field reversal. Therefore, I think the magnetic field dependence of thermopower and comparing it with magnetic field dependence of resistance may provide some extra information to understand the nature of the device. Would it be possible to present such data?

Response: We welcome referee's suggestion of measuring magneto thermopower. We wish to point that parallel field dependence of thermopower is already shown in supplementary information (Fig. S21). It shows that the thermopower is largely unaffected by the applied parallel magnetic field and thus unlikely to arise from an underlying spin/magnetic texture. Following referee's suggestion, we have now shown the low-field (perpendicular) magneto-thermopower data in Fig. R9 below. We find that the excess thermopower seems to diminish (more pronounced on h-doping) as we increase the magnetic field to 1 T and thus hints at the possibility of excess thermopower emerging due to orbital effects. Additionally, we plot a comparison of measured thermopower and that calculated using Mott formula at 1 T in Fig. R10, which shows a good agreement between the two curves at least on the h-side of doping. Thus, vanishing thermopower and a relatively better agreement with Mott formula in finite magnetic field suggest orbital nature of excess thermopower. However, it is to be noted that, we have not explored the thermopower behaviour in quantum hall regime as it is beyond the scope of the current manuscript and will be shown elsewhere. We have included this data in the supplementary information (Fig. S22) of the revised manuscript.

Fig. R9 Normalized thermopower S/S_{\max} as a function of n for different magnetic fields applied perpendicular to the plane of tBLG measured at 3 K.

Fig. R10 Comparison of measured thermopower and that calculated using Mott formula at fixed magnetic field of 1 T.

Changes in manuscript: Perpendicular field dependent thermopower data for device D3 has been presented in Supplementary information Fig.S22

References:

1. Polshyn, H., Yankowitz, M., Chen, S. *et al.* Large linear-in-temperature resistivity in twisted bilayer graphene. *Nat. Phys.* **15**, 1011–1016 (2019)
2. A. L. Sharpe, E. J. Fox, A. W. Barnard, J. Finney, K. Watanabe, T. Taniguchi, M. A. Kastner, D. Goldhaber-Gordon, Emergent ferromagnetism near three-quarters filling in twisted bilayer graphene. *Science* **365**, 605–608 (2019).
3. Arora, H.S., Polski, R., Zhang, Y. *et al.* Superconductivity in metallic twisted bilayer graphene stabilized by WSe₂. *Nature* **583**, 379–384 (2020).
4. Cao, Y. *et al.* Superlattice-induced insulating states and valley-protected orbits in twisted bilayer graphene. *Phys. Rev. Lett.* **117**, 116804 (2016).

REVIEWERS' COMMENTS

Reviewer #2 (Remarks to the Author):

I am satisfied with response to my questions.

The main result -- the unusual Seebeck coefficient close to $\nu=2$ -- seems to me a concrete nontrivial statement that does point out a new interesting facet of this system, that is of wide current interest. It also seems to me a robust enough statement . even though the overall characterization, noticed also by the other referee (and other referees including myself) is not fully controlled. The thermopower at $\nu=\pm 2$ behaves remarkably even if the numbers would be somewhat different. This finding is similar to what is being discussed in cuprates in Planckian regime, too, and will certainly motivate further work.

I agree with the response of the authors to the other referee concerning the sign of the effect at $\nu=2/-2$, which indeed could point to correlations. The overall particle-hole symmetry at is broken at large energy scales and the effect could persist to low energies in a correlated state and should indeed manifest itself in an opposite sign in the particle-hole symmetry related $\nu=2/-2$.

Reviewer #4 (Remarks to the Author):

Author provided constructive reply for all the reviewer's concerns. I think that the discussion of the manuscript is much clearer in the revised manuscript. This work provides novel thermoelectric phenomena near the magic angle graphene, and I agree that revised version is suitable for publication to the submitted journal.